# Mechanism of H_2_S in Inhibiting the Senescence and Browning of Fresh-Cut Potatoes

**DOI:** 10.3390/ijms26167785

**Published:** 2025-08-12

**Authors:** Zixu Lu, Nannan Liu, Wanjie Li, Lisheng Guan, Gaifang Yao, Hua Zhang, Kangdi Hu

**Affiliations:** 1School of Food and Biological Engineering, Hefei University of Technology, Hefei 230601, China; 2023111298@mail.hfut.edu.cn (Z.L.); 2023111322@mail.hfut.edu.cn (N.L.); 2024171636@mail.hfut.edu.cn (L.G.); yaogaifang@hfut.edu.cn (G.Y.); hzhanglab@hfut.edu.cn (H.Z.); 2Key Laboratory of Cell Proliferation and Regulation Biology, Ministry of Education, College of Life Science, Beijing Normal University, Beijing 100875, China; lwj@bnu.edu.cn

**Keywords:** hydrogen sulfide, fresh-cut potato, browning, antioxidant enzymes, reactive oxygen species

## Abstract

The market for fresh-cut fruits and vegetables is gradually expanding and is popular among consumers, but fresh-cut fruits and vegetables are highly susceptible to browning, causing a decrease in their quality and nutrition. Although anti-browning reagents and cryopreservation methods are often used for fresh-cut fruits and vegetables, the effects are not satisfactory. In this paper, hydrogen sulfide (H_2_S) donor NaHS solution was used for fumigation of fresh-cut potatoes to explore the mechanism of H_2_S signaling on the browning of fresh-cut potatoes at the biochemical level. Fresh-cut potatoes were fumigated with H_2_S and it was found that H_2_S treatment maintained better color compared with the browning of water control. Then, total phenolic content, reactive oxygen species-related metabolites hydrogen peroxide (H_2_O_2_) and superoxide anion (·O_2_^−^), along with malondialdehyde (MDA), the activities of antioxidant enzymes, and the browning-related enzymes polyphenol oxidase (PPO), catalase (CAT), peroxidase (POD), and phenylalanine amine lyase (PAL) were determined. The results of both principal component analysis (PCA) and correlation analyses consistently indicated that CAT activity showed a strong positive correlation with the browning degree of fresh-cut potatoes. The data indicated that H_2_S reduced the degree of browning, increased the total phenolic content, inhibited the accumulation of reactive oxygen species (ROS) content, inhibited POD, PPO, and PAL activities, and increased CAT activity.

## 1. Introduction

Potato (*Solanum tuberosum* L.), ranking as China’s fourth largest food crop, boasts the world’s largest cultivation area. This versatile crop serves multiple purposes as a staple food, vegetable, and animal feed, featuring an extensive industrial value chain that has earned it the reputation as the “second bread” [1,2]. As an annual herbaceous plant, potato is nutritionally valuable, containing abundant carbohydrates, proteins, vitamin C, essential amino acids, trace elements, and various bioactive compounds including phenolics, carotenoids, polyamines, and phytonutrients [3,4]. However, processing operations such as slicing and crushing induce enzymatic browning when potato tissues are exposed to air. This phenomenon significantly compromises product appearance and quality, posing substantial challenges to the potato industry [5,6,7,8]. The rapidly expanding fresh-cut produce market faces similar issues, where exposed cut surfaces accelerate oxidative deterioration and microbial decay, leading to a markedly reduced shelf life. The primary mechanism underlying fresh-cut potato browning involves antioxidant enzymes activity. In the presence of oxygen, antioxidant enzymes catalyze the oxidation of phenolic compounds to o-quinones, which subsequently polymerize with amino acids and proteins to form dark-colored melanoidins through enzymatic browning reactions [9,10]. Current anti-browning strategies, including chemical inhibitors, bioactive extracts, physical treatments, and biological approaches, have demonstrated limited effectiveness and present various drawbacks; the implementation of physical methods is often limited by their high costs, and chemical treatments, while more economical, raise potential food safety issues [11]. Consequently, developing more efficient and sustainable methods to control enzymatic browning in fresh-cut potatoes remains an important research priority. In recent years, there has been a surge in research exploring hydrogen sulfide’s (H_2_S) efficacy in delaying fruit browning across various species.

Hydrogen sulfide (H_2_S), recognized as the third gaseous signaling molecule in plants, plays crucial regulatory roles in diverse physiological processes including lateral root development, seed germination, stomatal movement, cellular autophagy, and programmed cell death [12,13,14,15]. As a potent reducing agent, H_2_S exhibits remarkable capacity to directly scavenge various reactive oxygen species (ROS), such as hydrogen peroxide, peroxides, hypochlorite, and peroxynitrite [16]. Antioxidant enzymes serve as the pivotal molecular defense system against ROS in living organisms, maintaining redox homeostasis through highly specific catalytic mechanisms. Antioxidant enzymes including polyphenol oxidase (PPO), catalase (CAT), peroxidase (POD), and phenylalanine amine lyase (PAL) collectively form the core component of cellular antioxidant networks by converting highly reactive ROS into less toxic products via cascade reactions. Beyond its antioxidant properties, H_2_S mitigates abiotic stress by reducing ROS accumulation while enhancing antioxidant enzyme activities and upregulating stress-responsive gene expression [17,18,19]. For instance, studies have documented H_2_S-mediated alleviation of chromium stress in cauliflower through antioxidant system reinforcement [20] and cold stress mitigation in peach fruits via ROS homeostasis regulation [21]. Recent advances have revealed H_2_S’s significant role in postharvest physiology. It effectively delays post-ripening softening, thereby extending the shelf life of fruits and vegetables [22,23,24]. Specifically, treatment with 1.6–3.2 mmol/L NaHS maintains antioxidant stability in strawberry fruits during storage at both ambient and low temperatures, resulting in prolonged postharvest quality [25]. Similarly, exogenous H_2_S application has been shown to effectively retard the ripening and senescence processes in postharvest tomato fruits [26,27,28].

Given the antioxidant potential of H_2_S, in the current study, we investigated the anti-browning effects of H_2_S treatment on fresh-cut potatoes. Using untreated fresh-cut potatoes as controls, we monitored the browning area during 0, 12, 24, 36, and 48 h-hour periods following H_2_S application. To elucidate the underlying mechanisms, we quantitatively analyzed ROS, including the metabolites hydrogen peroxide (H_2_O_2_) and superoxide anion (·O_2_^−^), along with malondialdehyde (MDA) content as an indicator of lipid peroxidation. Furthermore, the activities of key enzymes were assessed in the browning process, with these enzymes comprising the following: POD, CAT, PPO, and PAL. Based on these research findings, we propose that treatment with an appropriate concentration of H_2_S could delay the browning process of fresh-cut potatoes during storage.

## 2. Results

### 2.1. H_2_S Reduces Browning of Fresh-Cut Potatoes

Figure 1a presents the phenotype of fresh-cut potatoes treated with either H_2_S or distilled water (control) at 0, 12, 24, 36, and 48 h. Visual assessment revealed that while both groups exhibited progressive surface browning over time, the control group developed significantly more intense browning compared to the H_2_S-treated samples. Quantitative analysis of browning degree (Figure 1b) demonstrated distinct patterns between treatments. The H_2_S-treated group showed stable browning throughout the 48 h period, whereas the control group displayed a substantial increase. After 48 h, the control group’s browning intensity reached approximately three times that of the H_2_S-treated group, corroborating the visual observations in (Figure 1a), suggesting that H_2_S treatment effectively mitigates browning in fresh-cut potatoes. The changes in phenolic compounds, known for their antioxidant properties, are shown in Figure 1c. Both treatment groups exhibited increasing total phenolic content over time, ultimately reaching comparable levels. However, the H_2_S-treated group displayed a faster accumulation rate during the initial 24 h (0–24 h), followed by a slower increase in the subsequent period (24–48 h). Thus, the H_2_S-treated group maintained consistently higher phenolic levels throughout the experiment, suggesting that H_2_S preserves phenolic compounds as part of its anti-browning mechanism in fresh-cut potatoes.

### 2.2. H_2_S Inhibits the Accumulation of ROS Content in Potato

ROS serve as reliable indicators of oxidative stress in plant tissues. As shown in Figure 2a, H_2_S treatment significantly influenced H_2_O_2_ accumulation patterns in fresh-cut potatoes. Throughout the experimental period, H_2_S-treated samples consistently maintained lower H_2_O_2_ levels compared to the control group, although both treatments eventually converged to similar concentrations. The temporal dynamics of H_2_O_2_ accumulation revealed distinct patterns between treatments. From 0 to 24 h, both groups exhibited an initial increase followed by a decrease in H_2_O_2_ content, though these fluctuations were markedly attenuated in the H_2_S-treated group. Notably, at 12 h, the control group accumulated approximately 2.5-fold higher H_2_O_2_ than the H_2_S-treated samples. During the subsequent 24–48 h period, the H_2_S-treated group showed a characteristic decrease–increase pattern in H_2_O_2_ levels, while the control group maintained a steady upward trend. The oxidative stress difference peaked at 36 h, when control samples contained about 4.7 times more H_2_O_2_ than their H_2_S-treated counterparts. H_2_S suppressed H_2_O_2_ accumulation, indicating reduced oxidative browning.

Figure 2b illustrates the impact of H_2_S treatment on O_2_^−^ production rate in fresh-cut potatoes. Both H_2_S-treated and control (distilled water-treated) samples exhibited similar change patterns of ·O_2_^−^ production rate. However, the H_2_S-treated group consistently maintained lower O_2_^−^ levels throughout the experimental period. The maximum difference in ·O_2_^−^ accumulation occurred at 24 h, when control samples contained approximately 1.7-fold higher O_2_^−^ content compared to H_2_S-treated potatoes. This consistent suppression of O_2_^−^ generation by H_2_S treatment demonstrates its protective role against oxidative stress, thereby reducing browning in fresh-cut potatoes.

MDA, as the terminal product of membrane lipid peroxidation, serves as a reliable biomarker for assessing oxidative damage in plant tissues. As shown in Figure 2c, H_2_S treatment significantly influenced MDA accumulation patterns in fresh-cut potatoes. Throughout the experimental period, H_2_S-treated samples consistently exhibited lower MDA levels compared to the control group. In H_2_S-treated potatoes, MDA levels initially decreased during 0–12 h, followed by a substantial increase (12–24 h), and subsequently stabilized with a gradual decline. In contrast, control samples showed a continuous increase in MDA content during 0–24 h, with only a modest reduction occurring at 24–36 h before maintaining elevated levels thereafter. These findings clearly indicate that H_2_S treatment effectively suppresses MDA accumulation in fresh-cut potatoes, thereby mitigating membrane lipid peroxidation and subsequently delaying enzymatic browning processes.

### 2.3. Effect of H_2_S on Enzyme Activities of Potato

POD as a crucial oxidoreductase, playing a significant role in various physiological and biochemical processes in fruits and vegetables. As shown in Figure 3a, H_2_S treatment markedly influenced POD activity in fresh-cut potatoes. Compared to the control group, H_2_S-treated potatoes exhibited consistently lower POD activity throughout the storage period, with enzyme activity reaching its minimum (approximately 50% of the control) at 36 h. In contrast, the control group maintained relatively stable POD activity, showing no significant change by 48 h. These results demonstrate that H_2_S effectively suppresses POD activity, thereby delaying enzymatic browning in fresh-cut potatoes.

CAT plays a crucial role in enhancing antioxidative capacity by decomposing H_2_O_2_ in plant tissues. As shown in Figure 3b, H_2_S treatment significantly modulated CAT activity in fresh-cut potatoes. The H_2_S-treated group exhibited consistently higher CAT activity compared to the control, peaking at 36 h before gradually declining to levels comparable to the control group. The most pronounced difference between treatments occurred at 24 h, with the H_2_S-treated group showing 1.7-fold greater CAT activity than the control. These findings demonstrate that H_2_S effectively maintains elevated CAT activity, thereby enhancing H_2_O_2_ scavenging capacity and subsequently delaying enzymatic browning in fresh-cut potatoes.

PPO serves as the primary enzymatic catalyst responsible for tissue browning in higher plants. Figure 3c demonstrates the significant impact of H_2_S treatment on PPO activity in fresh-cut potatoes. Both control and H_2_S-treated samples exhibited an initial decline in PPO activity, reaching minima at 24 h simultaneously. Thereafter, PPO activity in both groups showed a gradual recovery. Notably, the H_2_S-treated group maintained consistently lower PPO activity throughout the experimental period. The most substantial difference was observed at 24 h, when control samples exhibited approximately 2.1-fold higher PPO activity compared to H_2_S-treated potatoes. These results clearly indicate that H_2_S treatment not only delays the onset of PPO activity resurgence but also effectively suppresses its subsequent increase, thereby maintaining PPO at significantly reduced levels and ultimately retarding the browning process in fresh-cut potatoes.

PAL, as a key enzyme in plant secondary metabolism and defense responses, showed distinct activity patterns under H_2_S treatment (Figure 3d). The H_2_S-treated group maintained consistently lower PAL activity throughout the experimental period, exhibiting only a gradual increase. In contrast, the control group displayed a rapid elevation in PAL activity during 0–24 h, followed by a sharp decline. The most significant difference between treatments occurred at 24 h, when control samples demonstrated 3.4-fold higher PAL activity compared to H_2_S-treated potatoes. These findings demonstrate that H_2_S effectively suppresses PAL activation and maintains its activity at reduced levels, thereby contributing to the retardation of enzymatic browning in fresh-cut potatoes.

### 2.4. Correlation Analysis and Principal Component Analysis

Based on browning, total phenol content, antioxidant enzyme activity, and the production rate of reactive oxygen species, correlation analysis and principal component analysis were conducted. As shown in Figure 4a, the correlation analysis was performed using Pearson’s correlation coefficient method; total phenol and reactive oxygen species content were positively correlated with browning, while antioxidant enzyme activity was negatively correlated with browning, among which CAT enzyme activity was highly negatively correlated with browning. As shown in Figure 4b, PC1 and PC2 contributed 99.94% and 0.06% of the data variation, respectively. PC1 represents the direction of maximum variance in the original dataset, typically capturing the most dominant pattern or trend in the data; PC2 represents a secondary principal component orthogonal to PC1, capturing the next most significant variation in the dataset. It can be seen that CAT showed the highest positive loading value in the PC1 direction, and POD showed the lowest positive loading value in the PC2 direction. It can be concluded that the change in CAT enzyme activity is significantly related to the degree of browning in potatoes.

## 3. Discussion

The global fruit and vegetable production sector is currently grappling with substantial postharvest preservation challenges, with browning and decay causing substantial economic losses and resource wastage [29]. Our study investigates the fundamental mechanisms underlying fresh-cut potato browning to establish a scientific basis for quality preservation. Recent studies have demonstrated H_2_S as an effective postharvest preservative for fresh-cut produce. In our experiments, H_2_S fumigation significantly reduced browning intensity in potato slices (Figure 1a,b) while maintaining elevated polyphenol levels (Figure 1c). These findings align with existing research showing H_2_S’s efficacy: (1) Leaf abscission rates progressively increased with elevating H_2_S concentrations (supplied as 0, 4, 8, 12, 16, 20, or 40 μL aliquots of 1 M NaHS solution) [30], and (2) vacuum infiltration with 1 and 2 mM H_2_S preserving phenolic compounds to prevent pericarp browning in lychee [31]. In addition, H_2_S was found to inhibit ethylene-induced petiole abscission in tomato [32] and regulate energy production to delay leaf senescence induced by drought stress in Arabidopsis [33], suggesting the role of H_2_S in delay plant senescence. The consistent anti-browning effects across diverse species suggest H_2_S’s potential as a universal postharvest treatment for browning mitigation.

Extensive research has established that postharvest browning and decay in fruits and vegetables are closely associated with ROS metabolism [34,35,36]. Excessive ROS accumulation induces cellular toxicity through membrane system damage and oxidative stress-mediated macromolecular degradation, ultimately accelerating tissue senescence and significantly reducing produce shelf life. This oxidative damage represents a fundamental challenge in postharvest preservation of horticultural crops. Our investigation of the anti-browning mechanism revealed that H_2_S treatment effectively suppressed ROS accumulation in fresh-cut potatoes, as evidenced by significantly lower levels of H_2_O_2_ and ·O_2_^−^ compared to controls (Figure 2). These findings corroborate previous observations in fresh-cut apples [37], suggesting a conserved ROS-regulation mechanism across different plant species. Furthermore, we evaluated MDA as a key biomarker of oxidative membrane damage. The H_2_S-treated group consistently maintained lower MDA content (Figure 2c), demonstrating reduced lipid peroxidation and better preservation of membrane integrity. These results collectively indicate that H_2_S-mediated browning inhibition operates through multiple protective mechanisms, including ROS scavenging and membrane stabilization.

POD catalyzes the oxidation of phenolic compounds using H_2_O_2_ as an electron acceptor, thereby promoting enzymatic browning [38]. Our results demonstrated significantly lower POD activity in H_2_S-treated potatoes compared to controls (Figure 3a), indicating effective suppression of this browning pathway. This finding aligns with previous reports in fresh-cut apples [39], suggesting a conserved mechanism of H_2_S action across different produce types. Plant antioxidant defense systems comprise two major components: enzymatic antioxidants including CAT and non-enzymatic antioxidants such as phenolic compounds [37]. CAT is a crucial antioxidant enzyme in plant cells that plays a vital role in cellular redox homeostasis by catalyzing the decomposition of H_2_O_2_ into water and oxygen [40], thereby protecting cells from oxidative damage. Our study revealed a significant negative correlation between CAT activity and browning degree, suggesting that higher CAT activity effectively inhibits browning progression. PCA further confirmed the dominant role of CAT in the browning process (Figure 4a,b). These findings demonstrate that CAT serves as a key regulator in controlling enzymatic browning through its antioxidant function, and our data revealed distinct patterns between treatments (Figure 2a and Figure 3b). While CAT activity in controls declined rapidly, H_2_S-treated samples maintained stable CAT levels, enabling efficient H_2_O_2_ scavenging. This preservation of CAT activity peaked at 36 h, coinciding with reduced browning incidence. The browning-related enzymes PPO and PAL showed complex temporal dynamics. PPO activity in H_2_S-treated potatoes exhibited an initial elevation (0–12 h) followed by rapid decline [41], ultimately remaining below control levels (Figure 3c). PAL activity, a key browning indicator, was consistently suppressed in treated samples during the critical 24 h period before converging with controls (Figure 3d). These results suggest that H_2_S modulates multiple enzymatic pathways to delay browning. Notably, while PPO activity was transiently higher in treated potatoes at 12 h (Figure 3c), overall browning remained significantly reduced (Figure 1b). This apparent discrepancy may reflect the multifactorial nature of browning, where factors beyond PPO activity—including lipoxygenase (LOX) activity, fatty acid composition, and membrane integrity—collectively determine browning susceptibility, as demonstrated in fresh-cut pears [42].

Based on the browning degree, total phenol content, enzyme activity, and reactive oxygen species production rate measured in this paper, PCA and correlation analysis were conducted. The high PC1 value obtained in the PCA indicates that CAT plays a major role in the browning process of potatoes (Figure 4b). Moreover, the results of the correlation analysis show that CAT has a significant negative correlation with browning (Figure 4a). This is because CAT has the ability to decompose H_2_S, thereby reducing the accumulation of reactive oxygen species and delaying the browning of potatoes.The stronger correlation between CAT activity and browning inhibition may be attributed to CAT’s specific catalytic capacity for H2O2 decomposition, which effectively scavenges ROS in plant tissues, thereby significantly delaying enzymatic browning. The PPO activity of potatoes temporarily increased at 12 h; these pronounced differences likely reflect the multifactorial nature of browning, where factors beyond PPO activity (including LOX activity, fatty acid composition, and membrane integrity) collectively determine browning susceptibility, as demonstrated in fresh-cut pears [40]. Browning is a complex physiological change process. The potential activation of alternative antioxidant enzyme pathways (e.g., LOX as observed in fresh-cut pears) by H2S treatment in potatoes represents an important avenue for further investigation.

## 4. Materials and Methods

### 4.1. Experimental Material

White-fleshed potatoes (*Solanum tuberosum* L.), originating from Anhui Province, were obtained from a wholesale market in Hefei, China.

### 4.2. Experimental Methods

#### 4.2.1. Sample Treatment

The potatoes were peeled and sliced to a thickness of 4 mm. Potato slices were divided into two groups: the treatment group (T) and the control group (CK). For the treatment group (T), a H_2_S donor solution containing 0.8 mmol/L sodium hydrosulfide (NaHS, 150 mL) was applied at room temperature in a sealed 3 L container. The control group (CK) received an equivalent volume (150 mL) of distilled water to ensure comparable experimental conditions. Samples from both groups were collected at 0, 12, 24, 36, and 48 h post-treatment. The collected samples were finely chopped, thoroughly homogenized in liquid nitrogen, and stored at −80 °C for subsequent analysis.

#### 4.2.2. Determination of Browning Degree and Total Phenol Content

To assess the browning degree, 3 g of each sample was homogenized with 6 mL of 0.2% (*w*/*v*) ascorbic acid (Vc) solution. Clarification was achieved through a second round of centrifugation under low-temperature conditions (4 °C). A blank control was prepared using 0.2% (*w*/*v*) ascorbic acid solution. The absorbance of the processed samples was measured at 420 nm using a spectrophotometer [43].

Total phenolic content was determined using a modified Folin–Ciocalteu method. Briefly, 2 g of potato sample was homogenized with 4 mL of 65% (*v*/*v*) acetone aqueous solution and incubated at 25 °C for 3 h. The homogenate was then centrifuged at 10,000× *g* for 30 min at 4 °C. For analysis, an aliquot of the supernatant was mixed with Folin–Ciocalteu phenol reagent and allowed to react for 10 min at room temperature. Subsequently, 0.5 mL of 10% (*w*/*v*) sodium carbonate solution was added, and the mixture was incubated at 25 °C for 2 h in the dark. The absorbance was measured at 765 nm using a spectrophotometer [44].

#### 4.2.3. Determination of ROS Content

The H_2_O_2_ content was determined using a colorimetric method based on a 4-(2-pyridinylazo)resorcinol (PAR) reaction. Briefly, 3 g of fresh tissue was homogenized with pre-chilled acetone (4 °C) at a 1:1 (*w*/*v*) tissue-to-solvent ratio. The homogenate was centrifuged at 3000× *g* for 20 min at 4 °C to obtain the supernatant containing the extracted H_2_O_2_. For quantification, the supernatant was reacted with 200 mM PAR solution, and the absorbance was measured at 508 nm using a spectrophotometer. A standard curve was prepared using known concentrations of H_2_O_2_ (0–200 μM) to calculate the H_2_O_2_ content in the samples, expressed as μmol H_2_O_2_ per gram fresh weight [45]

The O_2_^−^ content was determined using a hydroxylamine hydrochloride-based colorimetric method. For sample preparation, 3 g of fresh tissue was homogenized with 3 mL of ice-cold phosphate-buffered saline (PBS, 50 mM, pH 7.8) and centrifuged at 10,000× *g* for 30 min at 4 °C. The resulting supernatant containing ·O_2_^−^ was collected for subsequent analysis. For the colorimetric assay, 1 mL of supernatant was mixed with 1 mL of 10 mM hydroxylamine hydrochloride solution. After thorough mixing, 2 mL of 17 mM p-aminobenzenesulfonic acid and 2 mL of 7 mM α-naphthylamine solutions were sequentially added. The reaction mixture was incubated in a 30 °C water bath for 30 min before measuring the absorbance at 530 nm. A standard curve was established using sodium nitrite (NaNO_2_) solutions (0–50 μM) to quantify nitrite ion (NO_2_^−^) concentrations. The O_2_^−^ content was calculated based on the stoichiometric conversion of·O_2_^−^ to NO_2_^−^ and expressed as µmol ·O_2_^−^ per gram fresh weight [45].

The MDA content was determined using the thiobarbituric acid (TBA) reaction method. Briefly, 2 g of fresh tissue sample was homogenized with 5 mL of 5% (*w*/*v*) trichloroacetic acid (TCA) solution. The homogenate was centrifuged at 3000× *g* for 10 min at 4 °C to obtain the supernatant. For the assay, 2 mL of supernatant was mixed with 2 mL of 0.67% (*w*/*v*) TBA solution. The mixture was heated in a boiling water bath for 15 min, then immediately cooled on ice to terminate the reaction. The precipitate was removed by low-speed centrifugation at 4 °C. The absorbance of the resulting supernatant was measured at three wavelengths (450, 532, and 600 nm) using a spectrophotometer [46].

#### 4.2.4. Determination of Enzyme Activity

Fresh potato tuber tissue (3.0 g) was rapidly frozen in liquid nitrogen and pulverized to a fine powder. The frozen powder was immediately mixed with 3 mL of ice-cold extraction buffer to prevent protein denaturation. The homogenate was centrifuged at 12,000× *g* for 30 min at 4 °C. The resulting supernatant was carefully collected as the crude enzyme extract and maintained on ice for immediate use or stored at −80 °C for subsequent analyses.

The POD activity was determined spectrophotometrically using a guaiacol–H_2_O_2_ oxidation system. The reaction mixture contained the following: 25 mM guaiacol solution, 0.5 M H_2_O_2_ solution, and an appropriate volume of crude enzyme extract. The reaction was initiated by adding the enzyme extract to the pre-mixed substrate solution at 25 °C. The increase in absorbance at 470 nm was recorded every 10 s for 3 min using a spectrophotometer. One unit (U) of POD activity was defined as the amount of enzyme required to produce an absorbance change of 0.01 per minute under the assay conditions. The specific activity was expressed as units per milligram of protein (U/mg protein) or units per gram of fresh weight (U/g FW) [44].

The catalase activity was determined by monitoring H_2_O_2_ decomposition at 240 nm. The 3 mL reaction system contained the following: 2.89 mL of 50 mM phosphate buffer, 100 μL of crude enzyme extract, and 10 μL of 30% (*w*/*v*) H_2_O_2_ (final concentration 100 mM). The reaction was initiated by H_2_O_2_ addition at 25 °C. The decrease in absorbance at 240 nm was recorded every 10 s for 3 min using a UV-visible spectrophotometer. Catalase activity was calculated from the linear decrease in absorbance during the initial 60 s [47].

For the assay of PPO activity, the PPO activity was determined spectrophotometrically by monitoring catechol oxidation at 420 nm. The 3 mL reaction system contained the following: 2.8 mL of 50 mM phosphate buffer, 100 μL of 100 mM catechol, and 100 μL of crude enzyme extract. The reaction was initiated by enzyme addition at 25 °C. The increase in absorbance at 420 nm was recorded every 10 s for 3 min using a spectrophotometer. PPO activity was calculated from the linear portion of the reaction curve. One unit (U) of PPO activity was defined as the amount of enzyme causing an absorbance change of 0.01 per minute under the assay conditions [44].

The PAL activity was determined by monitoring the production of trans-cinnamic acid at 290 nm. The reaction system contained the following: 3 mL of 50 mM boric acid buffer, 0.5 mL of 20 mM L-phenylalanine, and 0.5 mL of crude enzyme extract. The reaction was initiated by enzyme addition and incubated at 25 °C. The increase in absorbance at 290 nm was recorded every 10 s for 10 min using a spectrophotometer. PAL activity was calculated from the linear portion of the reaction curve. One unit (U) of PAL activity was defined as the amount of enzyme required to produce an absorbance change of 0.01 per minute under the specified conditions, corresponding to the formation of 1 nmol trans-cinnamic acid per minute [44].

#### 4.2.5. Statistical Methods

The data were processed using SPSS Statistics 25 software (version 25.0; IBM Corp, Armonk, NY, USA), while experimental data visualization and analyses were conducted with Excel (Microsoft Office 2016; Microsoft Corp, Redmond, WA, USA). For correlation analysis and principal component analysis (PCA), the online tools provided by OmicShare (version 3.0, available at omicshare.com) were utilized, with access dated 16 June 2025. All experimental treatments were performed with three independent biological replicates (*n* = 3).

## 5. Conclusions

Based on the obtained results, we analyzed the browning of fresh-cut potatoes treated with 0.8 mmol H_2_S. Through the detection of total phenol content, reactive oxygen species generation rate, and antioxidant enzyme activity, it was found that H_2_S treatment had a significant browning-delaying function compared with water treatment. CAT enzyme activity has a high correlation with enzyme activity, and CAT enzyme activity plays an important role in delaying browning.

In the current study, our findings demonstrate that H_2_S treatment mitigates browning in fresh-cut potatoes through effective ROS scavenging, enhanced antioxidant enzyme activities, and coordinated regulation of browning-related enzymes. These mechanisms collectively maintain cellular redox homeostasis and delay quality deterioration in fresh-cut potatoes.

The present findings establish both a novel methodology and theoretical framework for potato shelf life extension. This H_2_S-based intervention demonstrates transferable potential for postharvest preservation of other fresh-cut vegetables, while warranting further investigation into its organoleptic impact on treated produce.

## Figures and Tables

**Figure 1 ijms-26-07785-f001:**
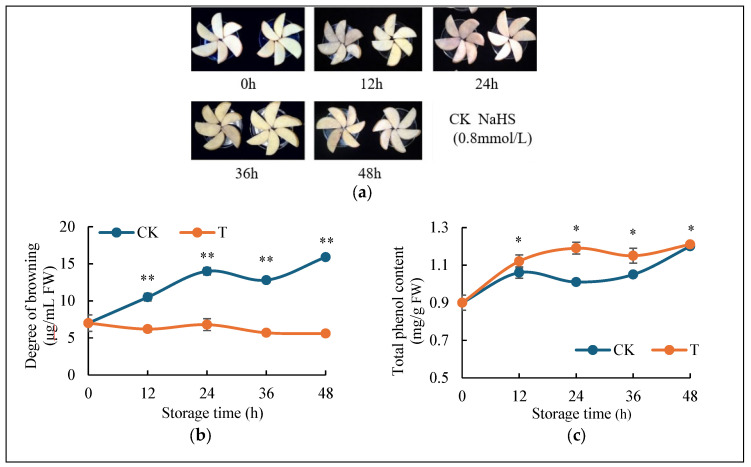
Effects of H_2_S on browning degree and total phenol content in fresh-cut potatoes. (**a**) Visual changes in fresh-cut potatoes after fumigation with H_2_S and distilled water for 0, 12, 24, 36, and 48 h. (**b**) Time–course effects (0–48 h) of H_2_S and control treatments on browning. (**c**) Time–course effects (0–48 h) of H_2_S and control treatments on total phenolic content. Data are means of three biological replicates ± standard deviation (SD). The symbols * and ** stand for *p* < 0.05 and *p* < 0.01 by *t*-test.

**Figure 2 ijms-26-07785-f002:**
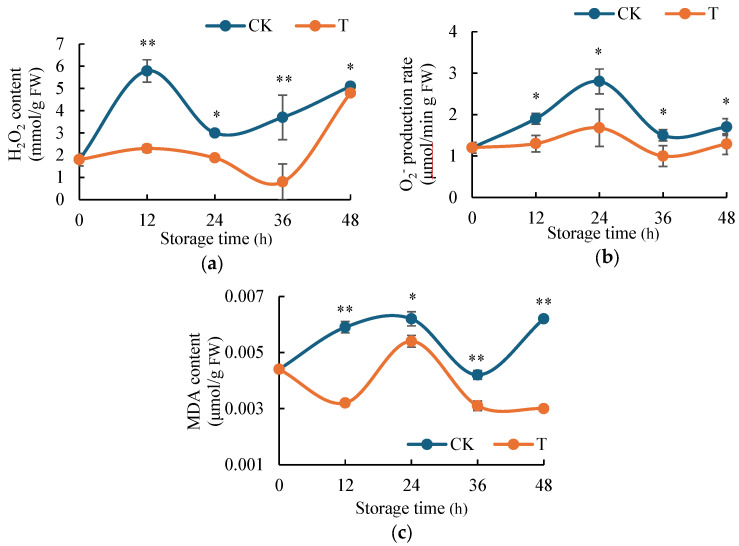
Effect of H_2_S on ROS content in fresh-cut potatoes. (**a**) H_2_O_2_ content variations in response to H_2_S fumigation were assessed over a 48 h period at specified intervals. (**b**) ·O_2_^−^ contents of fresh-cut potatoes after fumigation with H_2_S and distilled water for 0, 12, 24, 36, and 48 h. (**c**) MDA contents in H_2_S-fumigated and water-treated fresh-cut potatoes at designated time intervals (0–48 h). Data are means of three biological replicates ± standard deviation (SD). The symbols * and ** stand for *p* < 0.05 and *p* < 0.01 by *t*-test.

**Figure 3 ijms-26-07785-f003:**
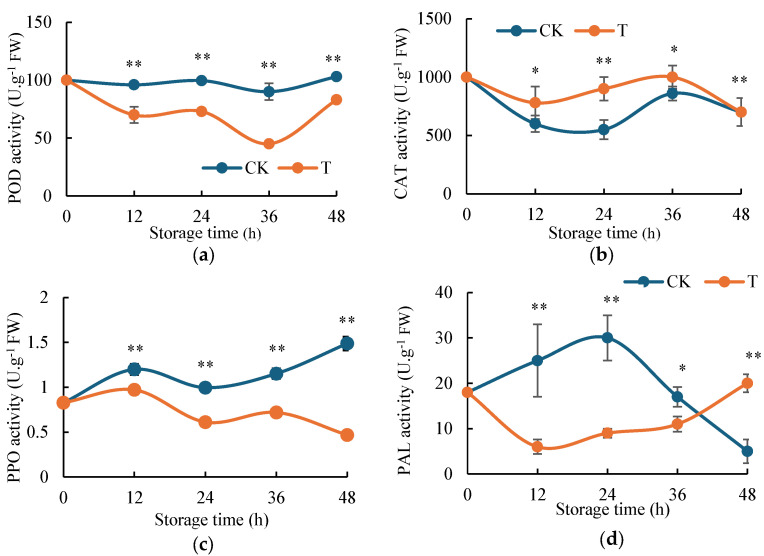
Effect of H_2_S in fresh-cut potato enzyme activity. (**a**) Changes in enzyme activities of POD after fumigation of fresh-cut potatoes with H_2_S and distilled water for 0, 12, 24, 36, and 48 h. (**b**) Changes in enzyme activities of CAT after fumigation of fresh-cut potatoes with H_2_S and distilled water for 0, 12, 24, 36, and 48 h. (**c**) Changes in enzyme activities of PPO after fumigation of fresh-cut potatoes with H_2_S and distilled water for 0, 12, 24, 36, and 48 h. (**d**) Changes in enzyme activities of PAL after fumigation of fresh-cut potatoes with H_2_S and distilled water for 0, 12, 24, 36, and 48 h. Data are means of three biological replicates ± standard deviation (SD). The symbols * and ** stand for *p* < 0.05 and *p* < 0.01 by *t*-test.

**Figure 4 ijms-26-07785-f004:**
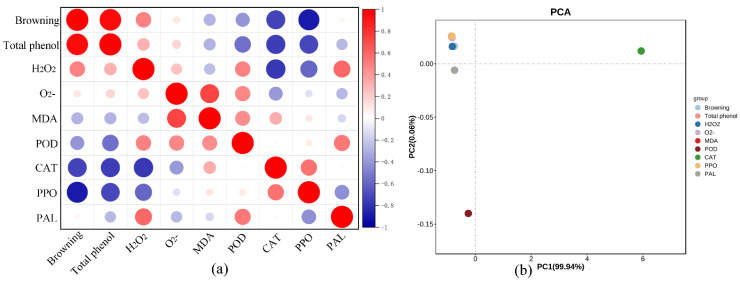
PCA of browning with enzyme activities and ROS in potatoes. (**a**) Correlation analysis of potato browning with ROS and enzyme activities. (**b**) PCA of potato browning with ROS and enzyme activities.

## Data Availability

Data were based on three biological replicates in each experiment, and the experiments were repeated independently three times. The statistical analysis of the data was based on Student’s *t*-tests at *p* < 0.05.

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
