# Peer review of "Mechanism of H2S in Inhibiting the Senescence and Browning of Fresh-Cut Potatoes"

_ijms, 2025, doi:10.3390/ijms26167785_

Round 1

Reviewer 1 Report

Comments and Suggestions for Authors

The comments are appended in the word file.

Comments on the Quality of English Language

It should be improved as detailed in the comments file.

Reviewer 2 Report

Comments and Suggestions for Authors

The theme of this work, although not so original, is interesting, especially because of the findings obtained in fresh-cut potatoes. Furthermore, it is a clearly written manuscript, the research proposal is clearly founded and supported with relevant and up-to-date references. The methodology is also clear and the techniques for determining variables are reliable and relevant. The results and corresponding figures are presented correctly, making them easy to understand. The discussion of results is acceptable and the conclusions are concrete and supported by the results obtained. The results obtained provide evidence that H2S treatment mitigates  browning in fresh-cut potatoes trough efective ROS scavening, enhanced antioxidant enzyme activities, and coordinated  regulation of browning related enzymes. However, there are some observations made below that would be worth the authors' consideration.

At the beginning of the paragraph of the section 3.1 H2S reduces browning of fresh-cut potatoes, it says: “Figure 1A presents the…”. I think it should say: “Figure 1 a presents the…”.

On line 6 from the same paragraph of the section 3.1 it says: “…of browning degree (Figure 1B) demonstrated…”. I think it should say: ““…of browning degree (Figure 1b) demonstrated…”.

On line 11 from the same paragraph of the section 3.1 it says: “… are shown in Fig. 1C. Both…”. I think it should say: “… are shown in Fig. 1c. Both…”.

Similar errors in the citation of figures are observed in the following sections 3.2, 3.3, 3.4 and in Discussion section 4.

The titles or labels of figures 1, 2 and 3 state the following: “… of fresh potatoes after fumigation with H2S and distilled water…”. I consider that, according to what is indicated in the materials and methods section, it should say: “… of fresh potatoes after inmertion in H2S or distilled wáter…”. In the section 2.2.1. Sample treatment, on lines 3-4 it says: “…a H2S donor solution containing 0.8 mmol/L sodium hydrsulfide (NaHS, 150 mL) was applied at room temperatura in a sealed 3 L container.

On lines 3-5 of the third paragraph of the section 3.3. Effect of H2S on enzyme activities of potato, it says: “…Both control and H2S-treated samples exhibited an initial decline in PPO activity, reaching their respective minima at different time points (12 h for control versus 24 h for H2S-treated).”. However, what is observed in Figure 3 c is somewhat different. In Figure 3c, an initial increase in PPO enzyme activity is observed for both groups. Furthermore, for both groups it is observed that the minimum activity of this enzyme occurred at the same time (24 h) of storage.     

Reviewer 3 Report

Comments and Suggestions for Authors

The manuscript is of great theoretical importance for the storage of vegetable and fruit crops

Unfortunately, the authors tested only one concentration of H2S donor solution containing 0.8 mmol/L sodium hydrosulfide (NaHS, 150 mL).

Maybe authors try to use higher concentrations of H2S donor solution containing sodium hydrosulfide and this will increase the storage of freshly cut vegetables and fruits in the future

Find my comments in the manuscript.

Round 2

Reviewer 1 Report

Comments and Suggestions for Authors

The correctons made by the authors accepted.

Author Response

The reviewers have not raised any additional comments.

Reviewer 3 Report

Comments and Suggestions for Authors
  1. Page  2 "...peroxidase (CAT)..." should be replaced with "сatalase (СAT)..."
  2.  Section "Conclusions" (Page 10-11). Please delete the sentence " Phenotypic observations and index determinations were conducted at five time points of 0, 12, 24, 36, and 48 hours after treatment."

Author Response

Thank you for your feedback. I have revised the manuscript accordingly as per your suggestions. Please refer to the attached file for the specific changes made.
